# Extracellular Vesicle-Derived Protein File from Peripheral Blood Predicts Immune-Related Adverse Events in Gastric Cancer Patients Receiving Immunotherapy

**DOI:** 10.3390/cancers14174167

**Published:** 2022-08-28

**Authors:** Fangli Jiang, Zhening Zhang, Xiaoyi Chong, Lin Shen, Meng Fan, Xuan Liu, Jin An, Zhi Peng, Cheng Zhang

**Affiliations:** 1Department of Gastrointestinal Oncology, Key Laboratory of Carcinogenesis and Translational Research (Ministry of Education/Beijing), Peking University Cancer Hospital & Institute, 52 Fucheng Road, Hai-Dian District, Beijing 100142, China; 2EVbio Technology Co., Ltd., Beijing 102200, China

**Keywords:** immune checkpoint inhibitors, immune-related adverse events, exosomes, ICOS, IDO1

## Abstract

**Simple Summary:**

Most gastric cancer (GC) patients have already benefited from immune checkpoint inhibitors, but some of them may terminate immunotherapy due to immune-related adverse events (irAEs). Extracellular vesicles have been shown to carry proteins, nucleic acids and other biomacromolecules to recipient cells, which is very important for exploring the potential of biomarkers of irAEs via EV-derived proteins. In 62 GC patients, EV-ICOS and EV-IDO1 were screened from 42 vital proteins as biomarkers of irAEs, and then confirmed in a validating cohort of 40 GC patients. In summary, EV-ICOS and EV-IDO1 can perfectly predict irAEs of ICI treated GC patients.

**Abstract:**

Immune checkpoint inhibitors (ICIs) initiate a new stage for gastric cancer (GC) therapeutics, and plenty of patients have already benefited from ICIs. Liquid biopsy promotes the development of precision medicine of GC. However, due to the lack of precision biomarkers of immune-related adverse events (irAEs), the safety of ICIs-treated GC patients cannot be guaranteed. In our study, GC patients treated with ICIs were included for investigating the correlation between irAEs of ICIs and corresponding outcomes. We also explored the potential of biomarkers of irAEs via EV-derived proteins. Dynamic plasma was taken from 102 ICIs-treated GC patients generated retrospectively or prospectively, who were divided into discovery and validating cohorts. Plasma EV-derived protein profiles were described, and two EV-proteins, inducible T-cell co-stimulator (EV-ICOS) and indoleamine 2,3-dioxygenase 1(EV-IDO1), from 42 vital proteins were screened to predict the prognosis of ICIs with irAEs. Our work is the first to propose that EV-proteins can predict ICIs-corresponding irAEs, which can be conducive to the diagnosis and treatment of GC patients, and to facilitate the screening of beneficiaries.

## 1. Introduction

Gastric cancer (GC) is the fifth most frequently diagnosed cancer and the third leading cause of cancer death worldwide [1]. With the rapid development of immunotherapy [2,3], clinical trials such as checkmate-649 and Attraction-2 have proved the efficacy of immune checkpoint inhibitors (ICIs) in the treatment of gastric cancer [4,5]. As proved by a series of studies, advanced GC patients harboring PD-L1-positivity/high combined positive score (CPS), microsatellite instability (MSI-H)/mismatch repair deficient (dMMR) and Epstein–Barr virus (EBV) infection are more likely to benefit from immunotherapy [6,7,8]. However, the clinical application of ICIs is frequently accompanied with adverse events, such as dermatological side effects, endocrine abnormality, hematopoietic abnormality, pneumonia and encephalitis. High grade irAEs occur in approximately half of patients receiving combinational ICI therapy (such as anti-CTLA-4 combined with anti-PD-1/anti-PD-L1) and around a quarter of patients receiving ICI monotherapy [9]. The manifestations of these immune related adverse events (irAE) are unpredictable. In addition, severe irAEs could be fatal and serve as a major cause of ICIs discontinuation [10,11]. Regrettably, it remains challenging for clinicians to properly and timely diagnose these events. Thus, it is an urgent need to develop potential biomarkers to predict irAEs.

Extracellular vesicles (EVs), defined as nano-size particles with a small diameter (40–160 nm), are secreted by almost all types of cells. Containing various bioactive molecules including proteins, lipids, and nucleic acids, EVs not only mediate signal transduction in cellular communication but also participate in the physiological process including immune regulation and cancer progression [12]. Due to these characteristics, EVs could be a perfect mode of liquid biopsy, as reported by substantial evidence focusing on its clinical application in tumor diagnosis and prognosis [13,14]. However, no study has reported that EVs are associated with irAEs.

In this study, a total of 102 GC patients who received ICI-based therapy were enrolled in our study and were divided into two independent cohorts. EVs derived from patient plasma were analyzed by protein microarray array paneled with 42 key proteins, and 2 vital members were screened from them to predict the prognosis and monitor the irAEs of GC patients treated with ICIs. Our study first described the potential of EV-derived proteins as novel biomarkers in predicting and monitoring irAEs, which offers reference value in the clinical practice.

## 2. Materials and Methods

### 2.1. Patient Information

Recruitment, treatment, management, follow-up and blood samples of all patients were executed by the Department of Gastrointestinal Oncology, Peking University Cancer Hospital & Institute. A total of 102 GC patients diagnosed with unresectable or advanced GC (including both GEJ (gastroesophageal junction) and non-GEJ types) and administrated with ICI-based regimens from August 2016 to April 2021 were enrolled. All patients had normal function of the heart, liver, lung, kidney and had been ruled out bacterial and viral infections. All patients’ irAEs were assessed following Management of Immune Checkpoint Inhibitor-Related Toxicity that was published in 2020 and updated to the NCCN Guidelines [15], and the definition and severity of irAEs were determined according to clinical examinations, biological and imaging data, the response rate was determined by the Response Evaluation Criteria in Solid Tumors (RECIST 1.1 Standards) [16]. Written informed consents was obtained from all donors. All specimens and relevant clinical information were approved by the Institutional Ethics Committee, Peking University Cancer Hospital & Institute for research. This study was conducted in accordance with the Declaration of Helsinki.

### 2.2. Specimen Preparation and Plasma EVs Isolation

Approximately 10 mL peripheral blood was collected with EDTA tube, centrifugated at 4 °C, 3000× *g*, 10 min for separating plasma and blood cells, then centrifugated at 25 °C, 1800× *g*, 8 min for obtaining lymphocytes after lysis red blood cells. Extracted blood components were stored at −80 °C for future experiments. EV was isolated from plasma samples by using the SEC-based qEV column (Izon Science, Christchurch, New Zealand), and the centrifugation method (CP100NX, Hitachi, Tokyo, Japan) was also applied in this work for EVs extracting. Finally, EV eluates were filtrated with an Amicon Ultra-4 10 kDa centrifugal filter device (Merck Millipore, Burlington, MA, USA).

### 2.3. EV Expression Array Determination and Identification

In our study, 42 candidate marker proteins, including 19 representative cellular biomarkers in tumor-immune microenvironment, 11 immune checkpoint modulators, 12 therapeutic biomarkers or targets of gastrointestinal cancer, and 4 canonical EV markers, were embraced in the EV expression array. Then the 46 key EV-proteins were printed onto a 3D modified slide surface (Capital Biochip Corp., Beijing, China) using Arrayjet microarrayer (Roslin, UK) to construct a protein chip spectrum. For EV-proteins capture experiment, the protein chip spectrum was initially blocked with 3% BSA (*w*/*v*) diluted in PBS for 1 h at room temperature, then incubated with 10 μL unpurified plasma sample that diluted (1:10) in wash buffer (0.05% Tween20 in PBS) sequentially at room temperature on a shake flask at 30 rpm for 2 h, and then incubated overnight at 4 °C. The next day, the slides were washed and incubated with biotinylated detection antibodies (anti-human-CD9, -CD63 and -CD81, LifeSpan BioSciences, Seattle, WA, USA) diluted 1:1500 in wash buffer. After that, Cy3-labelled streptavidin (Life Technologies, Carlsbad, CA, USA) was added in the slides for incubation 1 h at room temperature. After washing, the slides were scanned using the GenePix 4000A microarray scanner (Molecular Devices, San Jose, CA, USA) and GenePix Pro image analysis software (Molecular Devices, CA, USA) was used to analyze the signal intensity of fluorescent images.

### 2.4. Statistical Methods Database

Databases were downloaded from the Gene Expression Omnibus (GEO) (https://www.ncbi.nlm.nih.gov/geo/ (accessed on 2 November 2021)) and The Cancer Genome Atlas (TCGA) (https://portal.gdc.cancer.gov/ (accessed on 4 November 2021)), as well as gastric cancer cohort (*n* = 102) previously published by our group.

Wilcoxon–Mann–Whitney tests were used to analyze expressional diversity among subgroups. Clinical parameters were compared by the Chi-square test. Survival proportions of the irAEs group and non-irAEs group were evaluated by Kaplan–Meier analysis paired with Log-rank test. Only *p* value < 0.05 was considered statistically significant. All analyses and graphing were performed using SPSS 21.0 software or GraphPad Prism version 6.0.

## 3. Results

### 3.1. irAEs May Represent a Better Disease Control Rate of Immunotherapy in GC

As shown in the workflow chart (Figure 1), 102 GC patients received ICIs-based therapy were enrolled as a 62-patient discovery cohort and a 40-patient validating cohort in our work. In the discovery cohort, 149 time-scaled specimens were collected from 62 GC patients, including 62 specimens from BL (baseline), 32 specimens from CR (complete response)/PR (partial response), 20 specimens from SD (stable disease), and 35 specimens from PD (progressive disease). For all patients, 40.3% (25/62) received ICI monotherapy, 50% (31/62) received ICI combined with chemotherapy, 9.7% (6/62) received double-ICI therapy, while irAEs were observed in 56.5% (35/62) patients. The 35 irAEs cases were composed of 80% (28/35) level 1, 17.1% (6/35) level 2 and 2.9% (1/35) level 3. Among all 35 irAEs cases, 40% (14/35) were related with skin, 28.6% (10/35) with digestive system, 14.3% (5/35) with hematopoietic system, 8.6% (3/35) with endocrine system, 2.9% (1/35) with respiratory system, 2.9% (1/35) with kidney and 2.9% (1/35) with bone and muscle (Table 1).

IrAEs’ relevance with major clinical parameters were compared by chi-square test. Despite insignificant statistical relevance, all patients received double-ICI therapy developed irAEs. This was in accordance with previous reports, potentially due to double-ICI’s hypo-activation of anti-cancer immune responses remarkably igniting irAEs (Table 2). Compared to the patients without irAEs, patients diagnosed with grade ≥ 1 irAEs displayed comparable ORR (objective response rate) and better DCR (disease control rate) (Appendix A). Furthermore, patients who developed grade ≥ 2 irAEs displayed slightly longer, yet statistically insignificant irPFS/irOS (immunotherapy-related progression-free survival/overall survival) than patients without irAEs or with grade ≥ 1 irAEs (Appendix A) [17]. These basic features show that the occurrence of irAEs may correspond to a better disease control rate of immunotherapy in GC patients.

### 3.2. The Expressional Spectrum of Plasma EV-Proteins Identified ICOS and IDO1 as irAE Predictors

As described in the Materials and Methods section, pre-treatment plasma samples from the discovery cohort were assessed through an antibody-sandwiched protein array to generate the expression profiles of 42 key EV proteins and 4 canonical EV markers (Figure 2a). We then screened EV proteins associated with irAEs. Among the 42 EV proteins, ICOS and IDO1 displayed higher expression in GC without irAEs than in GC with irAEs (Figure 2b). The onset, stop, and duration of irAEs of six selected cases are exhibited (Appendix A). Cases 1–3 have the highest expression of EV-derived ICOS/IDO1, while cases 4–6 have the lowest expression of EV-derived ICOS/IDO1. Patient 1 developed grade 1 diarrhea at week 11 and was alleviated after receiving loperamide. Patient 2 developed grade 2 hypothyroidism at week 11 and took levothyroxine as replacement therapy permanently. Patient 3 developed grade 2 interstitial pneumonitis at week 15 and was cured after a one-month course of methylprednisolone. Patient 4–6 had an onset of grade 1 rashes at week 2, week 3, and week 5, respectively. Patient 5 and patient 6 also suffered from fever and anorexia, and their symptoms gradually improved after receiving nonsteroidal anti-inflammatory drugs. Moreover, ICOS and IDO1 displayed different expression profiles regarding the organs/systems related with irAEs (Figure 2c). These findings illustrated that EV-derived ICOS and IDO1 may provide clues for the irAEs rather than to predict ICI prognosis in ICI-treated GC patients.

### 3.3. EV-ICOS and EV-IDO1 Dynamically Predicted irAEs

In order to further explore whether EV-ICOS and EV-IDO1 can dynamically predict the occurrence of irAEs, time-scaled blood samples of GC patients were tracked and collected after baseline therapy. The content of CA199 and CA72.4, two prevalently applied circulating tumor markers in GC, displayed a more rapid reduction in grade irAEs ≥ 1 patients than irAEs = 0 patients along with the treatment (Figure 3a). Similarly, the expression values of EV-ICOS and EV-IDO1 were lower in patients with irAEs than in patients without irAEs across dynamic specimens from baseline to the six-month time point after treatment (Figure 3b). Notably, the time gap from treatment initiation to the occurrence of irAEs was slightly longer in patients expressing high baseline ICOS than in patients expressing low baseline ICOS (3.88 vs. 3.44 weeks), as well as in patients expressing high baseline IDO1 than in patients expressing low baseline IDO1 (4 vs. 3.4 weeks) (Figure 3c), indicating that lower baseline expression of EV-derived ICOS/IDO1 is related to earlier development of irAEs.

### 3.4. EV-Derived ICOS/IDO1 Were Irrelevant to Immunotherapeutic Effectiveness, Yet Refected Short-Term Changes of Circulating Tumor Marker CA72.4

We then analyzed the correlation between EV-derived ICOS/IDO1 and immunotherapeutic efficacy. The expression levels of EV-derived ICOS/IDO1 were comparable between patients with or without responses (Figure 4a). When stratifying patients with both irAEs and responses, we found that patients who reached an objective response while overcoming irAEs displayed higher EV-derived ICOS/IDO1 expression levels than other stratifications (Figure 4b), indicating that patients with higher EV-derived ICOS/IDO1 have a better chance to achieve both therapeutic effectiveness and safety. Both EV-derived ICOS and ICO1 were largely irrelevant to irPFS or irOS (Figure 4c). On the other hand, expressions of both EV-derived ICOS and IDO1 displayed a high correlation with the content of circulating tumor marker CA72.4 (Figure 4d). These data suggested that EV-derived ICOS and IDO1 were more likely to be associated with short-term treatment outcome.

### 3.5. EV-Derived ICOS/IDO1 Remained as Predictive Biomarkers for irAEs in Prospective Validation

To confirm the conclusion that we obtained in the discovery cohort, we then prospectively enrolled 40 ICI-treated patients and collected their pre-treatment plasma specimens as a validating cohort. Since all patients were alive during follow-up, only irPFS was recorded for analysis. For these 40 patients, 30% (12/40) received ICI monotherapy, 62.5% (25/40) received ICI plus chemotherapy and 7.5% (3/40) received double-ICI therapy, while irAEs were observed in 32.5% (13/40) patients. These irAEs were composed of 20% (8/40) grade 1, 10% (4/10) grade 2 and 2.5% (1/40) grade 3, while for all irAEs cases, 46.1 (6/13) were related with skin, 15.4% (2/13) with digestive system, 15.4% (2/13) with hematopoietic system, 15.4% (2/13) with endocrine system, and 7.7% (1/13) with bone and muscle (Table 3). When comparing irAE’s relevance with major clinical parameters, patients who received double-ICI therapy displayed a significantly higher trend to developed irAEs (Table 4), which was in accordance with the discovery cohort.

In the validating cohort, patients with irAEs achieved higher DCR than patients without irAEs (Appendix A), while patients who developed grade ≥ 2 irAEs also displayed slightly longer irPFS than patients without irAEs or with grade ≥ 1 irAEs (Appendix A). EV-derived ICOS and IDO1 displayed higher baseline expression in patients without irAEs than in patients with irAEs (Figure 5a), which was in accordance with our findings in the discovery cohort. However, the EV-derived ICOS/IDO1 expressional distributions among different organs or systems were distinct in validating cohort and discovery cohort (Figure 5b), suggesting that EV-derived ICOS/IDO1 were insufficient to predict the organs/systems-specific irAEs. Similar to discovery cohort, the time gap from treatment baseline to irAE occurrence was longer in patients expressing high baseline ICOS than in patients expressing low baseline ICOS (7 vs. 2.25 weeks), as well as in patients expressing high baseline IDO1 than in patients expressing low baseline IDO1 (8.25 vs. 2.22 weeks) (Figure 5c). Taken together, these results demonstrated that GC patients bearing higher EV-ICOS or EV-IDO1 had both lower risk and shorter interval to develop irAEs, representing a group of patients with better tolerance to ICIs.

Similar to discovery cohort, there is no evident correlation between the two EV-proteins and ORR/DCR (Figure 6a), while irAEs patients who reached objective response or disease control displayed higher EV-derived ICOS/IDO1 expression than other stratifications (Figure 6b), indicating that patients bearing high EV-ICOS or EV-IDO1 may achieve better therapeutic responses and less toxicities from ICIs. Additionally, levels of EV-derived ICOS/IDO1 were also positively corelated with the value of plasma CA72.4 (Figure 6c). Collectively, our work demonstrated that although EV-derived ICOS and IDO1 were largely irrelevant to immunotherapeutic responses or prognosis, they were efficient predictive biomarkers of irAEs. Dynamic alternations of these two EV-derived proteins in plasma provide guidance on clinical management of irAEs.

### 3.6. ICOS and IDO1 Were Related with Unique Tumor-Microenvironmental Features

Since EV-derived ICOS and IDO1 exhibited similar predictive roles for GC irAEs, we assessed their expressional profiles and found that these two EV-derived proteins were tightly linked in both discovery and validating cohorts (Figure 7a). The interaction of two molecules was also observed on tissue-derived transcript levels extracted from gastric cancer datasets TCGA and GSE62254 (Figure 7b), suggesting that ICOS and IDO1 may exert analogous functions in modulating tumor immune microenvironment (TIME). We next analyzed the TIME cell compositions related with ICOS and IDO1 in TCGA and GSE62254. In both datasets, higher expressions of ICOS/IDO1 on tissue level consistently displayed higher infiltrating immune cells, including B cells, T cells, NK cells, dendritic cells, macrophages, neutrophils and their related subtypes (Figure 7c,d). These phenomena reflected that ICOS and IDO1 expression in both plasma and tissue levels was reflective of unique tumor-microenvironmental features. However, due to the complexity of the TIME, ICOS/IDO’s mechanistic linkage with irAEs should be explored by further validations.

## 4. Discussion

Contemporarily, immunotherapy has become a central pillar of cancer treatment in various tumor types, including gastric cancer. Although a great many GC patients benefit from immunotherapy, the accompanying irAEs also disturb clinicians for they impede patients from deriving durable responses. IrAEs are a spectrum of diseases featured with unrestrained autoimmunity and inflammations affecting multiple organs. Though generally manageable, certain varieties of irAEs could be life-threatening. Therefore, it remains challenging to seek predictive biomarkers identifying patients at stake in irAEs to facilitate timely clinical intervention.

A series of studies have reported candidate biomarkers correlated with IrAEs. Gene expression profiles, C-reactive proteins, human leukocyte antigens (HLAs) and gut microbiome were found to be capable of predicting irAEs [10,18,19,20,21,22,23]. On serological level, higher or lower expression of specific genes, cytokines/chemokines and autoimmune antibodies were correlated with the occurrence of irAEs [24,25,26]. It has also been reported that early B cell changes, activated CD4 memory T cell abundance and TCR diversity were associated with irAE development [27,28]. It has been suggested MiRNAs, lncRNAs, and proteins carried by Evs are involved in carcinogenesis and the efficacy of ICIs [29], yet the involvement of extracellular vesicles with irAEs has not yet been elucidated. Our study for the first time reported the correlation between EV-derived proteins ICOS/IDO1 and irAEs in patients with GC.

ICOS and IDO1 are proteins related to immunotherapy; relevant target drugs have been produced [30,31], and several studies support that they were associated with allergic diseases [32]. It has been reported that activation of ICOS^+^ Tregs mediates allergic airway diseases, and targeting the ICOS/ICOS-L pathway may disrupt T follicular helper cell responses and ameliorate allergic asthma by disrupting the disease [33,34]. In addition, IDO1, as an immunosuppressive tolerogenic enzyme, has been reported as biomarkers of immune system activation and childhood allergic diseases [35,36]. As a T cell costimulatory receptor, ICOS has a series of immunoagonist antibodies that could multiply the immune response of T cells and improve the ICIs’ efficacy [37]. Moreover, IDO1 plays an important role in tumor immunosuppression, and multiple inhibitors have been evaluated In clinical trials [38,39]. However, we did not find proof that EV-derived ICOS or IDO1 exert an influence on immunotherapeutic responses or prognosis. Thus, our current work did not support the assumption of combining ICIs with ICOS-or IDO1-targeted agents.

## 5. Conclusions

Immunotherapy innovates cancer therapy while unpredictable irAEs restrict its clinical practice. In our study, a protein chip spectrum that contains 46 key tumor immune microenvironment molecules and classic targets was invented to screen irAEs related markers. Considering that plasma samples are easily obtained, less consumed, and minimally invasive, our detecting procedure was very convenient and accurate to predict irAEs. In conclusion, our results indicate that EV-ICOS and EV-IDO1 are efficient predictors of irAEs, which could assist the decision-making for clinicians and could reduce unnecessary suffering for GC patients treated with ICIs.

## Figures and Tables

**Figure 1 cancers-14-04167-f001:**
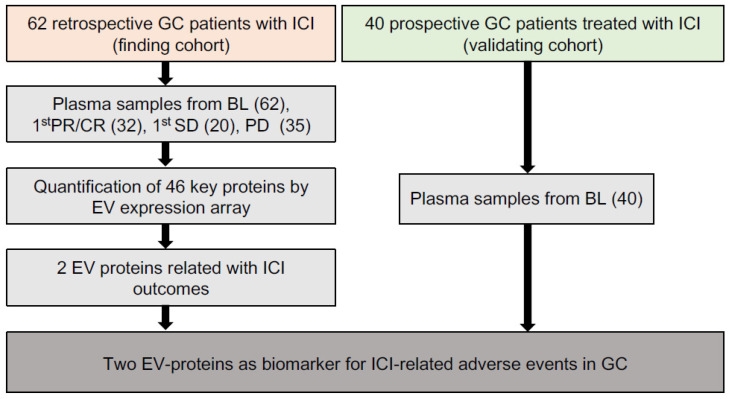
The overall workflow of our study.

**Figure 2 cancers-14-04167-f002:**
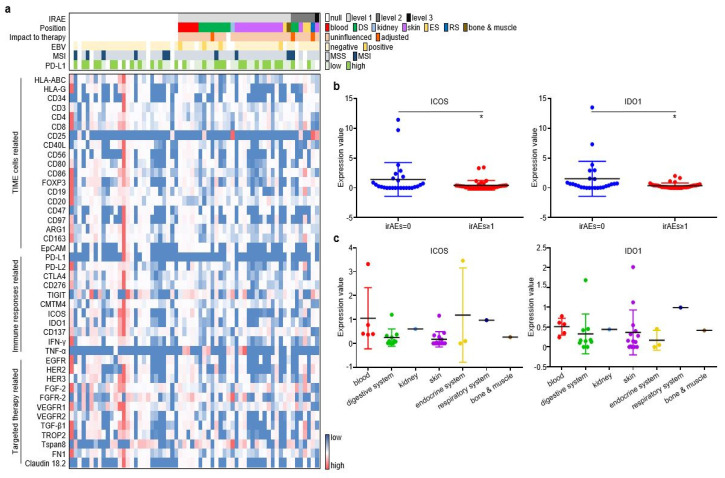
ICOS and IDO1 were screened to predict the occurrence of irAEs in 62 ICI-treated GC patients. (**a**) EV-proteins heat map in the discovery cohort. (**b**) EV-ICOS and EV-IDO1 were screened to monitor irAEs. (**c**) The expression degree of EV-ICOS and EV-IDO1 in different organs of GC patients with irAEs. (* *p* < 0.05, 1, ns: non-significant).

**Figure 3 cancers-14-04167-f003:**
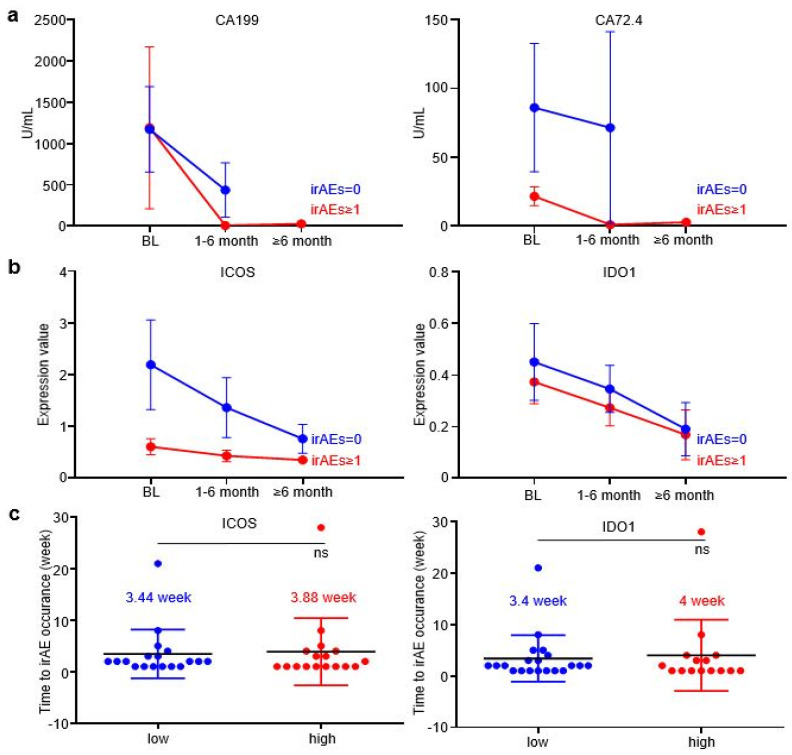
EV-ICOS and EV-IDO1 as dynamic predictors to forecast irAEs. (**a**) The changes of CA199 and CA72.4 in grade≥ 1 irAEs and without irAEs patients along with the treatment. (**b**) Dynamic changes of EV-ICOS and EV-IDO1 in patients with or without irAEs group after baseline. (**c**) The relationship between baseline level of EV-ICOS and EV-IDO1 with the occurrence speed of irAEs. ns, non-significant.

**Figure 4 cancers-14-04167-f004:**
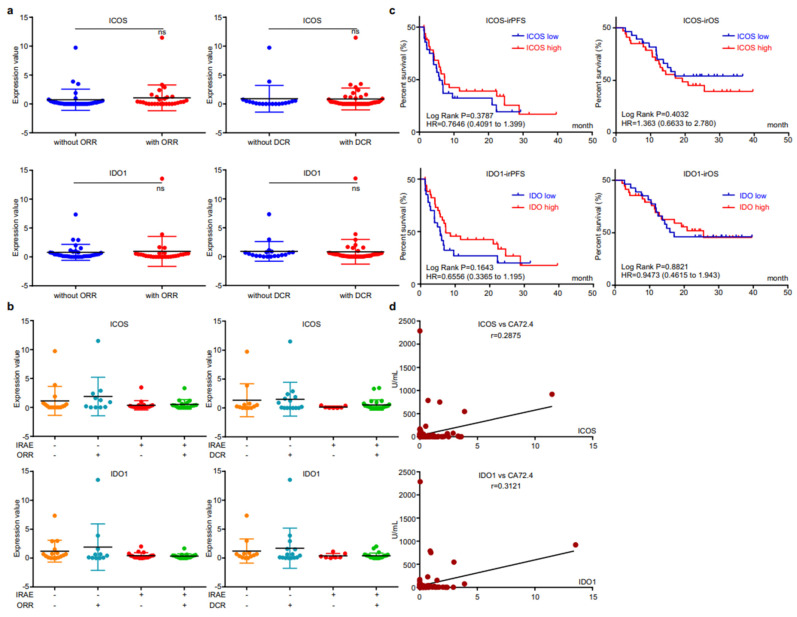
EV-ICOS/IDO1 were irrelevant to immunotherapeutic effectiveness. (**a**) Correlation between EV-ICOS/IDO1 and ORR/DCR. (**b**) The higher value of EV-ICOS/IDO1 suggests a better chance to achieve both therapeutic effectiveness and good safety. (**c**) Relevance of EV-ICOS/IDO1 and irPFS/irOS. (**d**) Relationship between EV-ICOS/IDO1 and the content of circulating tumor marker CA72.4. Different colors represent distinct IRAE and DCR status. ns, non-significant.

**Figure 5 cancers-14-04167-f005:**
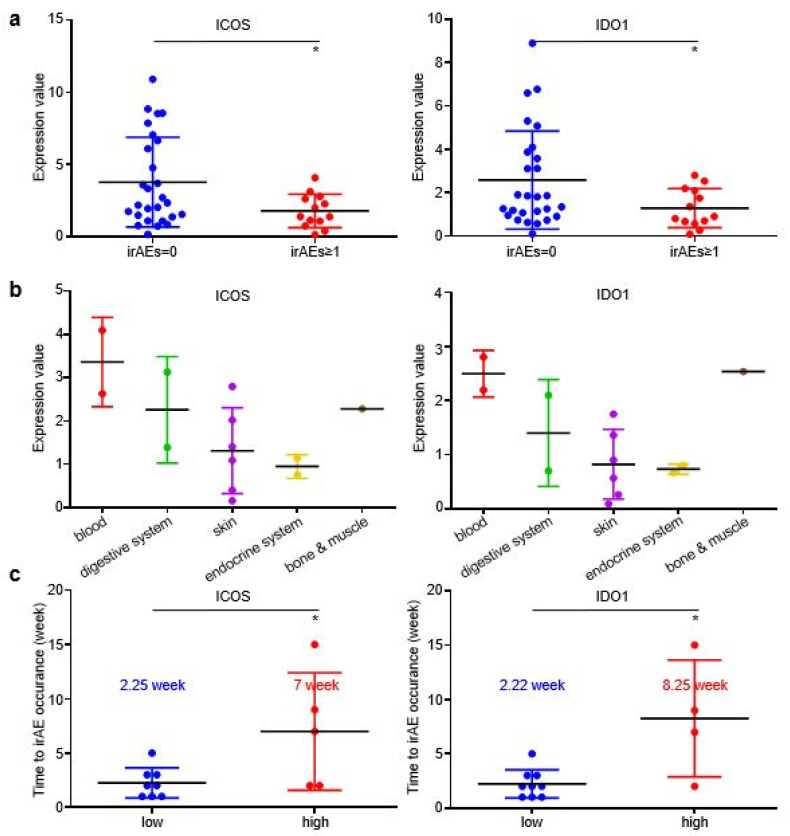
EV-ICOS/IDO1 remained as predictors for irAEs in validating cohort. (**a**) The baseline expression value of EV-ICOS/IDO1 was lower in irAEs groups. (**b**) EV-ICOS/IDO1 had no correlation with the distribution of irAEs organs/systems. (**c**) Patients with high ICOS/IDO1 had a later onset of irAEs. (* *p* < 0.05).

**Figure 6 cancers-14-04167-f006:**
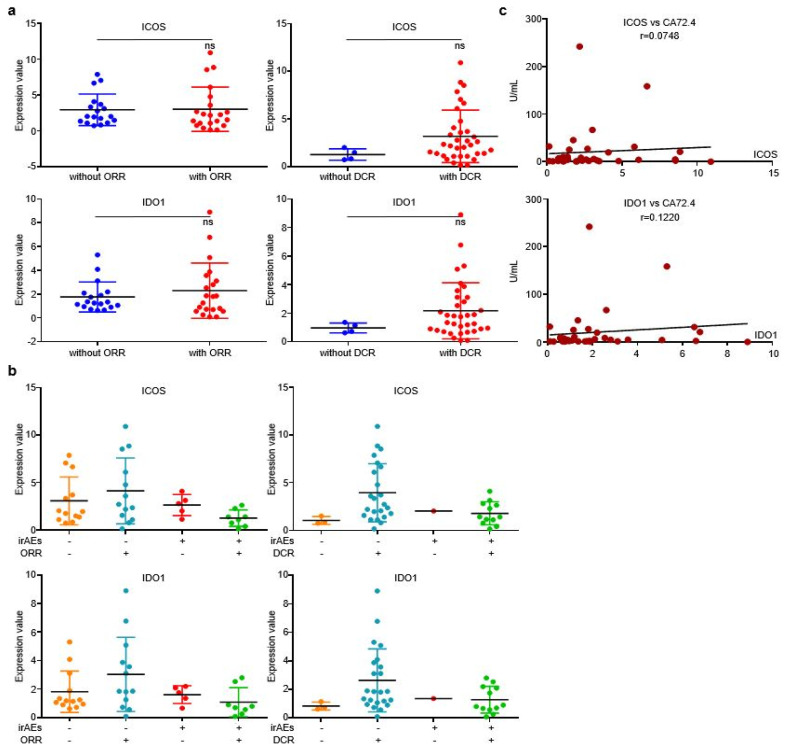
EV-ICOS/IDO1 was largely irrelevant to immunotherapeutic effectiveness in validating cohort. (**a**) EV-ICOS/IDO1 had no correlation with ORR/DCR. (**b**) The higher values of EV-ICOS/IDO1 predict the lower occurrence of irAEs in ORR/DCR reached patients. (**c**) EV-ICOS/IDO1 had a strong correlation with CA72.4. Different colors represent distinct irAEs and DCR status. ns, non-significant.

**Figure 7 cancers-14-04167-f007:**
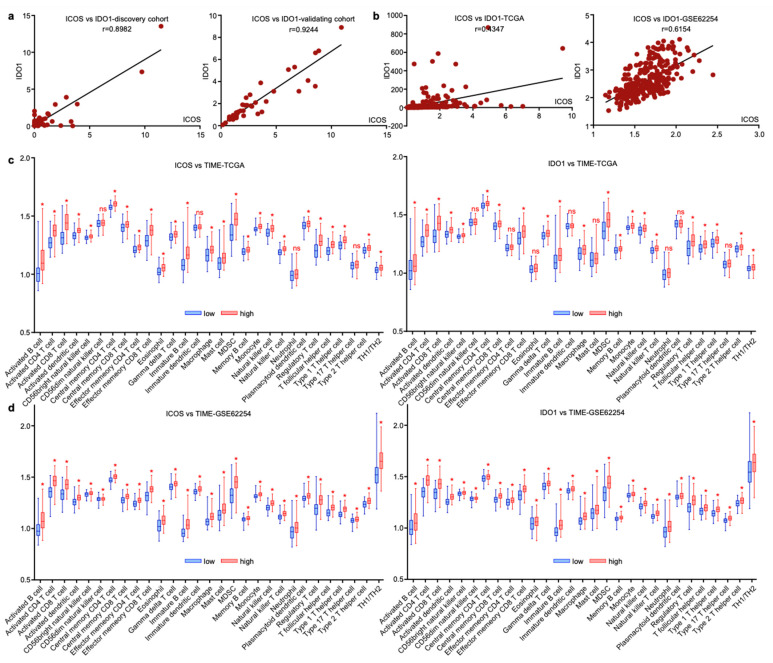
ICOS and IDO1 were related with unique tumor-microenvironmental features. (**a**,**b**) ICOS and IDO1 were positively correlated in our cohort and in TCGA and GSE62254 databases. (**c**,**d**) Cellular components of tumor immune microenvironment associated with ICOS and IDO1 in TCGA and GSE62254. (* *p* < 0.05, ns, non-significant).

**Table 1 cancers-14-04167-t001:** Clinicopathological features of the discovery cohort (n = 62).

Clinicopathological Features	Case Number (%)
Age (n = 62)	
≤65	39 (62.9)
>65	23 (37.1)
Gender (n = 62)	
Male	46 (74.2)
Female	16 (25.8)
Lauren’s classification (n = 62)	
intestinal type	27 (43.5)
diffuse type	13 (21)
mixed type	13 (21)
NA	9 (14.5)
TNM stage (n = 62)	
III	9 (14.5)
IV	53 (85.5)
Therapeutic line (n = 62)	
1st	36 (58.1)
2nd	11 (17.7)
≥3rd	15 (24.2)
Regimen (n = 62)	
ICI	25 (40.3)
ICI + chemotherapy	31 (50)
Double-ICI	6 (9.7)
HER2 status (n = 62)	
Neg	54 (87.1)
Pos	3 (4.8)
NA	5 (8.1)
MMR status (n = 62)	
MSS	45 (72.6)
MSI	10 (16.1)
NA	7 (11.3)
PD-L1 status (n = 62)	
Neg	21 (33.9)
Pos	27 (43.5)
NA	14 (22.6)
EBV status (n = 62)	
Neg	49 (79)
Pos	4 (6.5)
NA	9 (14.5)
irAE status (n = 62)	
0	27 (43.5)
1	28 (45.2)
2	6 (9.6)
3	1 (1.6)
irAE organs (n = 35)	
skin	14 (40)
digestive system	10 (28.6)
hematopoietic system	5 (14.3)
endocrine system	3 (8.6)
respiratory system	1 (2.9)
kidney	1 (2.9)
bone and muscle	1 (2.9)

NA: Not available, Neg: Negative, POS: Positive, PD-L1: Programmed cell death ligand 1, MMR: MisMatch repair, MSI: Microsatellite instability, MSS: Microsatellite stability; EBV: Epstein-Barr virus, ICI: Immune checkpoint inhibitor, TNM: Tumor Node Metastasis, HER-2: Human epidermal growth factor receptor 2.

**Table 2 cancers-14-04167-t002:** Correlation of irAE with clinicopathological features in discovery cohort (n = 62).

Clinicopathological Features	irAE Case Number (%)		*p* Value
	NO	YES	
total	27	35	
Age			0.7913
≤65	16 (59.3)	23 (65.7)	
>65	11 (40.7)	12 (34.3)	
Gender			1.0000
Male	20 (74.1)	26 (74.3)	
Female	7 (25.9)	9 (25.7)	
Lauren’s classification			0.2724
intestinal type	11 (40.7)	16 (45.7)	
diffuse type	7 (25.9)	6 (17.1)	
mixed type	3 (11.1)	10 (28.6)	
NA	6 (22.3)	3 (8.6)	
TNM stage			0.1604
III	6 (22.2)	3 (8.6)	
IV	21 (77.8)	32 (91.4)	
Therapeutic line			0.0608
1st	15 (55.6)	21 (60)	
2nd	8 (29.6)	3 (8.6)	
≥3rd	4 (14.8)	11 (31.4)	
Regimen			0.0676
ICI	13 (48.1)	12 (34.3)	
ICI + chemotherapy	14 (51.9)	17 (48.6)	
Double-ICI	0 (0.0)	6 (17.1)	
HER2 status			0.5590
Neg	21 (77.8)	33 (94.2)	
Pos	2 (7.4)	1 (2.9)	
NA	4 (14.8)	1 (2.9)	
MMR status			0.1747
MSS	16 (59.2)	29 (82.9)	
MSI	6 (22.3)	4 (11.4)	
NA	5 (18.5)	2 (5.7)	
PD-L1 status			0.2369
Neg	6 (22.3)	15 (42.9)	
Pos	13 (48.1)	14 (40)	
NA	8 (29.6)	6 (17.1)	
EBV status			0.1238
Neg	23 (85.2)	26 (74.3)	
Pos	0 (0)	4 (11.4)	
NA	4 (14.8)	5 (14.3)	

NA: Not available, Neg: Negative, POS: Positive, PD-L1: Programmed cell death ligand 1, MMR: MisMatch repair, MSI: Microsatellite instability, MSS: Microsatellite stability; EBV: Epstein-Barr virus, ICI: Immune checkpoint inhibitor, TNM: Tumor Node Metastasis, HER-2: Human epidermal growth factor receptor 2.

**Table 3 cancers-14-04167-t003:** Clinicopathological features of the validating cohort (n = 40).

Clinicopathological Features	Case Number (%)
Age (n = 40)	
≤65	20 (50)
>65	20 (50)
Gender (n = 40)	
Male	27 (67.5)
Female	13 (32.5)
Lauren’s classification (n = 40)	
intestinal type	14 (35)
diffuse type	6 (15)
mixed type	12 (30)
NA	8 (20)
Therapeutic line (n = 40)	
1st	25 (62.5)
2nd	8 (20)
≥3rd	7 (17.5)
Regimen (n = 40)	
ICI	12 (30)
ICI + chemotherapy	25 (62.5)
Double-ICI	3 (7.5)
HER2 status (n = 40)	
Neg	16 (40)
Pos	1 (2.5)
NA	23 (57.5)
MMR status (n = 40)	
MSS	31 (77.5)
MSI	8 (20)
NA	1 (2.5)
PD-L1 status (n = 40)	
Neg	7 (17.5)
Pos	24 (60)
NA	9 (22.5)
EBV status (n = 40)	
Neg	36 (90)
Pos	3 (7.5)
NA	1 (2.5)
irAE status (n = 40)	
0	27 (67.5)
1	8 (20)
2	4 (10)
3	1 (2.5)
irAE organs (n = 13)	
skin	6 (46.1)
digestive system	2 (15.4)
hematopoietic system	2 (15.4)
endocrine system	2 (15.4)
respiratory system	0 (0)
kidney	0 (0)
bone and muscle	1 (7.7)

NA: Not available, Neg: Negative, POS: Postive, PD-L1: Programmed cell death ligand 1, MMR: MisMatch repair, MSI: Microsatellite instability, MSS: Microsatellite stability; EBV: Epstein-Barr virus, TNM: Tumor Node Metastasis, HER-2: Human epidermal growth factor receptor 2.

**Table 4 cancers-14-04167-t004:** Correlation of irAE with clinicopathological features in validating cohort (n = 40).

Clinicopathological Features	irAE Case Number (%)		*p* Value
	NO	YES	
total	27	13	
Age			1.0000
≤65	14 (51.9)	6 (46.2)	
>65	13 (48.1)	7 (53.8)	
Gender			0.1569
Male	16 (59.3)	11 (84.6)	
Female	11 (40.7)	2 (15.4)	
Lauren’s classification			0.2033
intestinal type	11 (40.7)	3 (23.1)	
diffuse type	5 (18.5)	1 (7.7)	
mixed type	6 (22.3)	6 (46.1)	
NA	5 (18.5)	3 (23.1)	
Therapeutic line			0.3415
1st	15 (55.6)	10 (76.9)	
2nd	7 (25.9)	1 (7.7)	
≥3rd	5 (18.5)	2 (15.4)	
Regimen			0.0200
ICI	11 (40.7)	1 (7.7)	
ICI + chemotherapy	15 (55.6)	10 (76.9)	
Double-ICI	1 (3.7)	2 (15.4)	
HER2 status			1.0000
Neg	26 (96.3)	13 (100)	
Pos	1 (3.7)	0 (0)	
NA	0 (0)	0 (0)	
MMR status			0.2286
MSS	19 (70.4)	12 (92.3)	
MSI	7 (25.9)	1 (7.7)	
NA	1 (3.7)	0 (0)	
PD-L1 status			0.0152
Neg	18 (66.7)	3 (23.1)	
Pos	4 (14.8)	6 (46.2)	
NA	5 (18.5)	4 (30.7)	
EBV status			0.2532
Neg	25 (92.6)	11 (84.6)	
Pos	1 (3.7)	2 (15.4)	
NA	1 (3.7)	0 (0)	

NA: Not available, Neg: Negative, POS: Positive, PD-L1: Programmed cell death ligand 1, MMR: MisMatch repair, MSI: Microsatellite instability, MSS: Microsatellite stability; EBV: Epstein-Barr virus, ICI: Immune checkpoint inhibitor, TNM: Tumor Node Metastasis, HER-2: Human epidermal growth factor receptor 2.

## Data Availability

The data included in this study were described in Appendix A.

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
