# Peer review of "Extracellular Vesicle-Derived Protein File from Peripheral Blood Predicts Immune-Related Adverse Events in Gastric Cancer Patients Receiving Immunotherapy"

_cancers, 2022, doi:10.3390/cancers14174167_

Round 1
Reviewer 1 Report (Previous Reviewer 1)
The authors have largely addressed my comments in the revision.
Reviewer 2 Report (Previous Reviewer 2)
Now the quality of the manuscript is improved.
This manuscript is a resubmission of an earlier submission. The following is a list of the peer review reports and author responses from that submission.
Round 1
Reviewer 1 Report
Major comments:
- Fig. 2B and 2C – What test was used for significance? The data do not look convincing
- Missing evidence that EVs were indeed collected – Western blots needed (see MISEV2018 guidelines)
- Fig. 5C – Although the authors showed a significant difference, only a small sample size was used and the data were highly variable.
Minor comments:
- Line 47 – cause of CANCER death
- Use the term EV instead of exosome where appropriate (see MISEV2018 guidelines)
- Fig. 2B and 2C – using smaller dots instead of large circles would make the data clearer
- Line 217 – Replate 8tratifying with stratifying
Author Response
Major comments:
Fig. 2B and 2C – What test was used for significance? The data do not look convincing. Missing evidence that EVs were indeed collected – Western blots needed (see MISEV2018 guidelines)
Response: Thank you for your questions. In Fig. 2B and 2C, unpaired T test was used to compare the expression level of EV-ICOS and EV-IDO1 in patients with different irAEs status, as well as in patients with different organ damage status. The evidence of EVs is displayed in a previous work of our lab in figure supplementary 4 (doi:10.1002/jev2.12209), and the western blots are shown as follows.
Fig. 5C – Although the authors showed a significant difference, only a small sample size was used, and the data were highly variable.
Response: Thank you for your questions. In our work, one discovery cohort including 62 GC patients and one validating cohort containing 40 GC patients were collected. In validating cohort, only 13 patients occurred irAEs, so fig. 5C shows a small number of specimens.
Minor comments:
Line 47 – cause of CANCER death
Use the term EV instead of exosome where appropriate (see MISEV2018 guidelines)
Fig. 2B and 2C – using smaller dots instead of large circles would make the data clearer
Line 217 – Replate 8tratifying with stratifying
Response: We feel sorry for these careless mistakes. We have corrected the wording and have adjusted the figures accordingly.

Reviewer 2 Report
Dear Authors,
Extracellular vesicle-derived protein file from peripheral blood predicts immune-related adverse events in gastric cancer patients receiving immunotherapy by Jiang is an interesting manuscript. However, it needs further experimental evidence to make the manuscript stronger.
Comment 1. The authors discovered EV-ICOS and EV-IDO1 proteins from the pool of 42 vital proteins, which are highly relevant to Immune checkpoint inhibitors (ICIs) in GC patients. Is there any gender differences noted in the expression of EV-ICOS and EV-IDO1 genes? How many control patient’s samples were employed in the study?
Comment 2. EV is a cargo, do any microRNA or non-coding RNA-packed EVs in the immunotherapy patients plasma?
Comment 3. It will be nice to separate the gender, control vs immunotherapy treated patients for EV-ICOS and EV-IDO1 genes? This will help the readers to understand.
Author Response
Comment 1. The authors discovered EV-ICOS and EV-IDO1 proteins from the pool of 42 vital proteins, which are highly relevant to Immune checkpoint inhibitors (ICIs) in GC patients. Are there any gender differences noted in the expression of EV-ICOS and EV-IDO1 genes? How were many control patient’s samples employed in the study?
Response 1: Thank you for your precise questions. Actually, a total of 102 GC patients received ICIs-based therapies and were divided into two cohorts: a 62-patient discovery cohort and a 40-patient validating cohort. It should be noted that all patients enrolled in our study received immunotherapy. There was no design of placebo control since we aimed to explore the biomarkers of immune-related adverse events in this study. We thought that gender might not be an important affecting factor regarding irAEs. Besides, our subgroup analysis according to different gender also failed to indicate any statistical difference.
Fig S4. Gender differences of EV-ICOS and EV-IDO1 genes. (a,b) Gender differences of EV-ICOS and EV-IDO1 genes in discovery cohort. (c,d) Gender differences of EV-ICOS and EV-IDO1 genes in validating cohort.
Comment 2. EV is a cargo, do any microRNA or non-coding RNA-packed EVs in the immunotherapy patients’ plasma?
Response 2: Thank you for your pertinent questions. For sure, EVs have been shown to deliver proteins, microRNAs, noncoding RNAs and other biomacromolecules to recipient cells, and play an important role in the crosstalk between the host immunity and tumor cells. Here we list a few literatures which interrogate how plasma EV-derived miRNAs or circRNAs might potentially affect the efficacy of immunotherapy or mediate the immune resistance (doi:10.1136/jitc-2019-000376, doi:10.1136/jitc-2019-000376, doi: 10.1016/j.ymthe.2022.01.046). However, our current project focused only on the profile of EVs-encapsulated proteins, which means that we did not detect the expression of RNAs. Actually, we have identified a distinct EV-derived protein file to be correlated with the therapeutic outcome of immunotherapy in a previous pioneering work (doi:10.1002/jev2.12209). The current study, to our knowledge, is the first to investigate the correlation between EV-derived protein file and the incidence of irAEs among patients receiving immunotherapy.
Comment 3. It will be nice to separate the gender, control vs immunotherapy treated patients for EV-ICOS and EV-IDO1 genes? This will help the readers to understand.
Response 3: Thank you for your comment. This comment is as profound as the first comment. Actually, our main purpose was to investigate the biomarkers of irAEs in ICI-treated GC patients. In addition, the expression levels of EV-ICOS and EV-IDO1 were not significantly different across different genders, which indicates that EV-ICOS and EV-IDO1 are universal biomarkers.
